# Economic Evaluation of Mental Health Effects of Flooding Using Bayesian Networks

**DOI:** 10.3390/ijerph18147467

**Published:** 2021-07-13

**Authors:** Tabassom Sedighi, Liz Varga, Amin Hosseinian-Far, Alireza Daneshkhah

**Affiliations:** 1Centre for Environmental and Agricultural Informatics, School of Water, Energy and Environment (SWEE), Cranfield University, Cranfield MK43 0AL, UK; t.sedighi@cranfield.ac.uk; 2Department of Civil, Environmental and Geomatic Engineering, Faculty of Engineering, UCL, London WC1E 6BT, UK; l.varga@ucl.ac.uk; 3Centre for Sustainable Business Practices, University of Northampton, Northampton NN1 5PH, UK; amin.hosseinianfar@northampton.ac.uk; 4Research Centre for Computational Science and Mathematical Modelling, School of Computing, Electronics and Mathematics, Coventry University, Coventry CV1 5FB, UK

**Keywords:** Bayesian network, cost-effectiveness intervention, evaluation, flood risk management, mental health impacts, QALY

## Abstract

The appraisal of appropriate levels of investment for devising flooding mitigation and to support recovery interventions is a complex and challenging task. Evaluation must account for social, political, environmental and other conditions, such as flood state expectations and local priorities. The evaluation method should be able to quickly identify evolving investment needs as the incidence and magnitude of flood events continue to grow. Quantification is essential and must consider multiple direct and indirect effects on flood related outcomes. The method proposed is this study is a Bayesian network, which may be used ex-post for evaluation, but also ex-ante for future assessment, and near real-time for the reallocation of investment into interventions. The particular case we study is the effect of flood interventions upon mental health, which is a gap in current investment analyses. Natural events such as floods expose people to negative mental health disorders including anxiety, distress and post-traumatic stress disorder. Such outcomes can be mitigated or exacerbated not only by state funded interventions, but by individual and community skills and experience. Success is also dampened when vulnerable and previously exposed victims are affected. Current measures evaluate solely the effectiveness of interventions to reduce physical damage to people and assets. This paper contributes a design for a Bayesian network that exposes causal pathways and conditional probabilities between interventions and mental health outcomes as well as providing a tool that can readily indicate the level of investment needed in alternative interventions based on desired mental health outcomes.

## 1. Introduction

Natural hazards can have large societal impacts. It is estimated that they caused 7700 human fatalities and $110 billion loss of infrastructural assets worldwide just in 2014 [1]. Out of the set of natural hazards, flooding is often regarded as the most frequently occurring type of natural disaster with increasing risk to society (particularly in the UK and Europe), and with the greatest impact on humans [2]. Of the €150 bn in reported damages caused by natural hazards in Europe in the period from 1999 to 2009, over one-third of the damages (i.e., €50 bn) were due to flooding. Furthermore, annual flood losses are expected to increase five-fold by 2050 and nearly 17-fold by 2080 in Europe, drawing attention to the urgency for cities in Europe to construct resilience against flooding [3].

Similar to other natural disasters, when flooding occurs, it creates significant damage to homes, communities, businesses, public services, and so forth. Residential properties usually suffer the greatest proportion of flood damage, with 25% of total damage, for example, costs of £320 million incurred by 10,465 properties due to flooding [4]. Therefore, flood risk management is a disaster administration priority for European countries, particularly the UK.

It has been argued that flood risk management is usually measured as direct property and infrastructure losses, since these are the most important input for cost–benefit analysis to guide the government bodies to invest in flood risk management strategies [5]. However, the impacts of flooding on urban populations are multi-faceted and wide-ranging. It is well-known that floods also have enormous impacts on people, both directly and indirectly (see Figure 1). Distinctions must be made between direct/indirect and tangible/intangible flood damages. Direct damage becomes immediately visible in the affected areas due to close physical contact with floodwater, while indirect damage emerges with a time delay and/or outside the area affected by floods [6].

The most apparent intangible impact of flooding is on human health. Direct intangible damage is a primary loss, which manifests as physical injury or even loss of life. Indirect health impacts are mental health disorders, which are caused by the experience of being flooded, or being impacted during the restoration process. Estimating flooding impacts will provide valuable insights for decision making and risk mitigation, policy-making, civil protection, emergency alertness and response, insurance and reinsurance, damage estimation practice/research, and so forth [7].

On account of this, a comprehensive societal cost–benefit assessment must take into account the intangible losses caused by floods, such as psychological disorders or anxiety [8], as well as tangible losses. Due to the anticipated complications of converting intangible values, such losses are generally ignored in risk assessments [9]. Thus, the economic evaluation of the convincing levels of investment, which should be made into interventions to mitigate flood risk and support recovery from floods, is very challenging.

In this paper, the primary focus is on the evaluation of flood impacts on human health, particularly mental health [10]. Flood impact assessment is a key component of the practice of flood risk management. Flood risk is defined in the European Flood Directive as “the combination of the probability of a flood event and of the potential adverse impacts on human health, the environment, cultural heritage and economic activity associated with a flood event” [11]. Flood damage estimates are therefore useful at all the stages of what is known as the flood mitigation cycle.

It is thus crucial to embrace social, political, environmental and other conditions, such as flood likelihood and local priorities, in the comprehensive evaluation. The evaluation method must be also able to swiftly determine changed investment requirements as the incidence and magnitude of flood events continue to grow. This quantification is essential and must examine various direct and indirect flood impacts on flood related outcomes in a probabilistic manner.

In this paper, we illustrate how the Bayesian network (BN) probabilistic method can be used efficiently for ex-post economic evaluation, as well as ex-ante for future assessment, and indeed near real-time for reallocation of investment into interventions.

In general, BNs provide a robust and flexible analytic approach to the challenge of complex health datasets, which pose specific computational challenges because of missing data, large or small size of data, complexity (of relationships not only between variables but also within the datasets themselves), changing populations, and nonlinear relationships between exposures and outcomes [12]. Unlike the regression-based models or multivariate copula models [13], BNs are historically most commonly used in clinical risk prediction analysis and risk stratification [14] in medicine. They provide compact and instinctive graphical representations that can be used to conduct causal reasoning and risk prediction analysis. Furthermore, the cause and effect statements can be readily exploited in BN networks to reduce the computational time and cost. This can be considered another important advantage of this modelling approach in comparison to the conventional approaches, such as joint probability distribution, which only encodes the values of the outcomes of interest, given the input variables. Therefore, Bayesian networks offer a compact tool for dealing with the uncertainty and complexity of a system. In this study, the benefits, efficiency and limitations of the BN-based evaluation method will be studied by examining the effect of flood interventions upon mental health, which is a gap in current investment analyses.

In order to construct the proposed probabilistic methods, we need to have a holistic overview of the relationship between flood events, their aftermath and population well-being, and risk factors causing psychological disorders. The psychological health impacts of flooding, and their relationship with flooding and other risk factors will be briefly discussed in the next section. Estimating the cost of flooding on human health, in particular on human mental health, is very challenging but essential so that investment into interventions can be evaluated against reduced mental health impacts. It is essential to use metrics/methods to monetize mental health impacts.

In order to reduce the damage to the community and people caused by flood events, environmental agencies are using various interventions each with different outcomes, efficiencies and costs. Any combination of interventions results in different value for money, with multiple conditional dependencies between interventions, choices of implementation and their contexts. This study provides an efficient construction of a probabilistic BN that displays causal pathways and their probabilities between interventions and mental health outcomes, as well as providing a tool that can readily indicate the level of investment needed for alternative interventions based on anticipated mental health outcomes.

## 2. Psychological Impacts of Flooding

The psychological impacts of flooding can be very significant and long-lasting. Difficulties in evaluating the mental health impacts of flooding arise because accurate diagnosis of any condition is not straightforward, and mental health impacts are often under-reported and can be overlooked in comparison to the physical health impacts.

There are some studies that evaluate the impact of flooding on mental health. In one of the earlier studies, Reference [15] conducted a study to evaluate the psychological impacts attributed to severe flooding in Kentucky, US in 1984. The findings indicated that the flood exposure had psychological impacts on the population and these impacts included depression and anxiety. Ref. [16] conducted a similar study with a group of participants from a flood affected population in the town of Lewes in the UK, to evaluate both the physical and mental health effects of the flooding in the area in the year 2000. The study’s findings identified a high correlation between flood exposure and psychological distress. Such physical and psychological consequences denote people’s vulnerabilities as they interact with nature [17]. Tapsell et al. go on to assert that the quantification of natural disaster impacts on population health is an intricate task due to the delay in receiving feedback from the population. Nevertheless, they conducted a similar piece of research on the impacts of the flooding in 1998 in large parts of England and Wales. Their longitudinal study took place over a period of four and a half years, and evaluated both physical and psychological impacts. The top four psychological impacts in the few weeks or months after the flood were claimed to be ‘Anxiety’, ‘Increased Stress Levels’, ‘Sleeping Problems’, and ‘Mild Depression’. Nonetheless, the order by which these health effects were reported varied from one geographical area to another.

The UK and England in particular are prone to flooding. In 2005, a severe flood hit Carlisle, UK, and many homes were affected. Carroll et al. [18] conducted qualitative research to evaluate the psychological impact of this specific flood and to evaluate the impacts of disasters and how they could inform policies. They concluded that the main psychological impacts are anxiety, stress and post-traumatic stress disorder (PTSD). Another study reported specifically that females were psychologically more vulnerable than males in the event of flooding [19].

Research in other regions provide similar findings. Vietnam is also susceptible to natural disasters and specifically to flooding. Bich et al. [20] highlight that controlling communes significantly reduces psychological impacts when flooding occurs. There are different strategies to mitigate the impact of flooding from low impact development technologies [21] and relocation [8] to forestation [22]. It was also reported that relocation during flood recovery, as an intervention, is correlated with a 600% increase in mental health symptoms [8].

Zhong et al. [23] provide a better understanding of what is currently known regarding the long-term health impacts of flooding and the factors that may influence health outcomes (including psychological health) by conducting a systematic mapping. Their findings indicate that 68% of these studies focused on the psychological impacts of flooding, whereas only 16% of these studies evaluated the physical effects following exposure to flooding. They have underlined that future research needs to quantify the long-term health impacts of flooding and identify their major determinants using some novel quantitative tools. These tools should be able to quantify the influence of multiple social interventions, such as flood management, on long-term health outcomes, and also identify the most influencing factors affecting the psychological and physical impacts.

## 3. Cost Estimation of Flooding

Estimating the cost of flooding on human health, including mental health, is extremely challenging. The study by [24] reports the best indicative estimates for the loss of life and health for the 2015 to 2016 UK winter floods (i.e., £43 m, within a range of £32 m, £54 m). The best estimate of loss of life and health impacts is calculated as “surrogate cost of fatalities” plus “surrogate cost of health impacts”, where surrogate cost of fatalities (£5 m) is measured as the number of fatalities due to flooding times the ‘average value of prevention of fatality’. The “surrogate cost for health impacts” (£38 m) is calculated as ‘cost per household’ times the ‘number of households affected’. The first term (cost per household) is defined as household willingness to pay per year to avoid the health impacts of extreme flood events, times the discount factor in the year, and the second term (number of properties affected) is measured as the ‘number of residential properties flooded’ times the ‘number of households likely to have health affects’.

Most studies focused on direct impacts. The common types of health metrics used are: death; hospital admissions and out-patient visits; cases of acute morbidity or injuries; and mental disorders or reduction in well-being [25].

However, loss of life or the number of injured are commonly used to measure the health burden associated with any natural disaster and, therefore, the impact of flooding on individuals’ mental health is often overlooked. In order to monetize health impacts in the flooding context, the following should be considered: Healthcare resource use; Productivity loss; Dis-utility from suffering or life-shortening.

The monetary value of the latter component is typically evaluated by wealth–health trade-offs that the affected people reveal in surrogate markets, or which can be implemented through multiple choice experiments. The monetary value of dis-utility associated with an adverse health outcome is thus attributed to the willingness to pay (WTP) to avert outcomes or, when considering mortality risk, the value of a statistical life that is derived from individuals’ aggregated WTP for a small change in survival probabilities [25]. In the studies that used loss of life numbers to quantify health impacts, only a few of them applied a monetary value to this outcome by multiplying it with a value of statistical life. This is not surprising, given that monetizing death is less useful for descriptive studies that are investigating trends in effects, or for studies reporting results from population-based surveys.

Matsushima et al. [26] valued WTP to avoid mental damages from flooding using an option value approach, in order to address potential strategic bias that would lead to an over-valuation of WTP. The WTP was also reported in [27] to estimate the willingness to contribute in terms of labour, in order to circumvent the fact that most individuals would not be able to afford any financial payment. They have also concluded that flood damage was estimated on average to represent about 20% of households’ annual income. However, it was not possible to solve the welfare loss from morbidity and well-being reduction from the welfare loss due to damages to assets. Poor households were found to be more vulnerable to flooding as the associated damages made up a significantly larger portion of their annual income. Households heavily dependent on agricultural activities were also found to be more vulnerable.

The UK Environment Agency (EA) has recently studied the new economic costs for the mental health impacts of flooding by analysing the data provided by Public Health England (PHE). It was illustrated that the mental health prevalence of people disrupted or affected by flooding is considerably higher than the unaffected groups, over 12 and 24 month periods. The findings of the study are comparable to the results from the flooding occurred in 2007. It was also reported that the chance of any type of mental health outcomes among the affected population will increase with the flood severity (or depth of flood). The same study confirms that WTP could be a very useful metric to evaluate the social cost of the flood impact; however, it cannot be used to include the actual cost of the mental health outcomes to the economy.

A study commissioned by Defra suggested that households were, on average, willing to pay £200 per year (2004 prices) to avoid the negative health impacts of flooding (e.g., for events occurring less frequently than 1 in 75 years) [28]. Defra’s climate change risk assessment report [29] considers the costs of treating a case of mild depression following a flood event as £970 (2010 prices), which can be used as an indicator of mental health impacts. It should be noted that these monetary values are normally used as predictions in policy assessments to allocate resources to protect against an abstract individual’s loss of life or suffering from harm. They were not designed to include post-event analysis. Without any official post-event values, however, these values were used as a surrogate in both the 2007 and 2013 to 2014 ’cost of flood reports’ to provide an indicative sum for loss of life and health impacts. Nevertheless, the above research and other studies conducted by PHE intended to better understand the health impacts of flooding and these efforts have resulted in some changes in the 2013 to 2014 methodology for estimating the cost of flooding. More research is urgently required to estimate the cost of treating cases of anxiety, depression and PTSD, using the existing and other relevant data. The factors affecting the cost of mental health outcomes are:Knowledge of each outcome (or condition);Prevalence of these outcomes;Presence of known treatment plans;Duration of any treatment;Likely impact of the outcome over the short term in terms of days of work lost.

The quantification of the benefits of flood risk prevention measures is still an unresolved challenge in disaster management research. In particular, there is no clear flood risk management to quantify the effect of interventions in reducing the flooding impacts on people including the effects on the affected population’s mental health. The most widely adopted framework in flood risk reduction is represented as the calculation of the expected damages as a function of flood hazard, physical vulnerability and exposure [30]. According to this framework, flood hazard is characterized by specific return periods—an estimate of the likelihood of the flood. Moreover, together with the vulnerability, it is usually expressed as an index, while the exposure is expressed with the unit(s) of measurement of the elements at risk, in physical or monetary terms. However, floods can impact socio-ecological systems in various forms, and therefore this framework is limited to assess damages to constructed infrastructure only. Furthermore, there have been a few other attempts to provide such a holistic risk assessment (see [31,32]), yet, these methods primarily focus on assessing direct tangible costs, since there is only enough relevant information to justify decisions regarding structural risk reduction measures. The main challenge with traditional frameworks is that they neglect the fact that the magnitude of flooding costs is determined by the adaptive behaviour of communities to absorb the flood hazards. It is obvious that the human dimension of vulnerability must be addressed as one of the main elements of the flood risk. The human aspect of vulnerability relates to the ability to cope with the hazard after a flood and the capacity to adapt to the flood hazard before the event [33]. More recently, emphasis has shifted from just being prepared, informed and minimising the dimension of vulnerability, to strategic proactive planning and management. There are two main reasons for this shift in recent flood risk management:The uncertainty of flood occurrence has noticeably increased due to intensified climate change; andThe consequences of flooding considerably depend on the behaviour of the affected people and their capability to adapt.

## 4. Mitigating the Impact of Flood Health Damages

There are various interventions that reduce the damages caused by flood events on communities, local environmental agencies, with different outcomes, efficiencies and costs. For instance, an intervention can be to use an early warning system (EWS) to reduce the amount of direct tangible costs (e.g., people can move transportable properties outside of the exposed area when the flood hazard is anticipated). The aim of a flood warning system is to provide useful information, for instance, by issuing alerts or activating the required protection measures with a view to improve decision making and action. The connections and feedback between hydrological and social spheres of early warning systems are key elements of successful flood mitigation.

The behaviour of the public and first responders during flood situations is determined by their preparedness, and is heavily influenced by numerous behavioural traits such as the perceived benefits of protection measures, risk awareness, or even denial of the effects that might occur. In the UK, the Environment Agency (EA) has an important role in warning citizens about the risk of flooding with a view to reducing the impact of flooding from rivers and the sea as well as pluvial floods.

In November 2009, Cumbria experienced devastating flooding in its different regions due to the heaviest rainfall ever recorded in the UK [34]. Following this, the EA carried out qualitative and quantitative research to evaluate the impact of the EA’s flood intervention methods, including early warning systems, partnership work, and on-ground assistance. These research works also highlight opportunities to improve the EA’s ability to respond to future floods. The affected residents received warnings in the various forms, including EA Flood-line Warnings Direct, people’s own observations of the local area, warnings on weather forecasts, warnings from neighbours, friends, and/or family, the Flood-line, and warnings by the emergency services [35]. They found that early warning systems themselves could add to stress. Most people also found Flood Action Groups very helpful in protecting their homes against flooding. However, they valued the idea of making a flood action plan, though such flood action planning was not yet widespread.

Another intervention to reduce mental health harm is to relocate people away from the affected regions as soon as possible, and to support them during and after a flood. This also has rebound effects.

A probabilistic method is needed to consider all sources of uncertainty that may influence an intervention in a particular context in order to evaluate its value for money There are complex relationships between flood events, their aftermath, population well-being and risk factors causing people’s health deterioration and/or psychological disorders [36].

To understand the potential benefits or drawbacks of any intervention for reducing the damaging impacts on people’s health, and particularly their mental health, the nature of the hazard and the vulnerability of the community and its exposure need to be taken into consideration. Figure 2 illustrates a conceptual model of a customised version of the risk framework considering the impacts that EWS may have on people [33]. In this framework, Hazard refers to the potential occurrence of flooding that may cause loss of life, injury, or other health impacts, as well as damage and loss of property, infrastructure, livelihoods, service provision and environmental resources.

## 5. Flooding and Health Risk Factors: Modelling Approaches

There are currently several statistical methods to explore the relationship between flooding and the health risk factors discussed above. For instance, a multivariable logistic regression model was proposed by [37] to model individuals’ revealed changes in mental health outcomes between year one and year two after flooding, by considering some of the above-mentioned factors. A similar method (logistic regression analysis) was used to select the risk factors and to predict flooding victims’ mental health states [38]. Applications of the multivariate regression-based methods are very limited. Their performance is hugely dependent on the size of the dataset, and can hardly be used to efficiently model the complex relationships between flood events, their aftermath impacts, and risk factors causing people’s health deterioration and/or psychological disorders. In addition, they are not useful for assessing risks in complex systems and scenarios of ‘decision making under uncertainty’ to optimise cost-effective decisions.

Alternatively, probabilistic methods, particularly BNs, have become an increasingly popular method for modelling uncertain and complex systems [39] and are considered a powerful tool for presenting knowledge and interpreting insights from available data [40]. Applications of BN methods are found in a growing number of studies and disciplines [41]. BNs are particularly useful for evaluation due to their capability of classification based on observations. BNs have been also widely used in environmental management contexts and are appropriate for decision making under uncertainty [42,43]. Moreover, unsupervised learning from a dataset can be performed using a BN by adopting the learning algorithm to find both structure and conditional probabilities. This means the evaluator does not need to know how to create a BN, although it is possible to aid the learning algorithm with prior knowledge about relations and probabilities. Dealing with uncertainty when evaluating policy is a challenge that can be addressed using BNs, since uncertain probabilities of variables may be safely ignored to reach the desired probabilistic quantity of a random variable. Furthermore, BNs engage directly with subjective data in a transparent way. Hence, the method could be considered more as a tool to explore beliefs, evidence and their logical implications, rather than as a means to ’prove’ something in a somewhat absolute sense. They, therefore, are useful for producing the balanced judgements required for evaluation in a Value for Money context. Additionally, BNs can be used privately to structure and inform the evaluator’s understanding, or publicly in a participatory process to stimulate and challenge collective views [41]. Finally, BNs are user-friendly and practical, and can present the ‘story’ behind a finding intuitively and graphically.

## 6. Evaluation Method: Probabilistic Graphical Models

Bayesian network (BN) is a mathematical model that graphically and numerically represents the probabilistic relationships between random variables through the Bayes theorem. This technique is becoming popular for aiding decision-making in several domains due to the evolution of computational capacity, which makes possible the calculation of complex BN [44]. Applications of BN methods are found in a growing number of disciplines and policies [14,41,45,46].

In the BN, as a probabilistic graphical model which is used to represent knowledge about an uncertain domain [44], each random variable is represented by a node. The BN, B, is a directed acyclic graph (DAG) that represents a joint probability distribution over a set of random variables $X=(X_{1},X_{2},\ldots ,X_{n})$. The network is defined by the pair B={G,θ}, where G=(X,E) is a DAG with nodes X representing random variables and edges *E* representing the direct dependencies between these variables. θ is the set of probability functions (i.e., node probability table), which contains the parameter $\theta_{x_{i}|pa_{i}}=P_{B}\left(x_{i}|pa_{i}\right)$ for each $x_{i}$ in $X_{i}$ conditioned by the parent set of $x_{i}$, denoted by $pa_{i}$, as the set of parameters of $X_{i}$ in G. The joint probability distribution defined by B over X is given in Equation (Equation 1): (1)$$P_{B}\left(X_{1},\ldots ,X_{n}|\theta \right)=\prod_{i=1}^{n}P_{B}\left(X_{i}|pa_{i}\right)=\prod_{i=1}^{n}\theta_{X_{i}|pa_{i}}.$$

A simple example of a BN is illustrated in Figure 3, where the probability of a person having cancer can be computed in terms of “Relatives had cancer” ($Y_{1}$) and whether the person is smoking or not (denoted by $Y_{2}$).

As can be observed, a conditional probability table (CPT) is attached to each node. The CPT on each node is associated with the conditional probability distribution, as given in Equation (Equation 1). The CPTs (or conditional probabilities) can be estimated from the observed data or expert opinions [14,48]. A link (or ‘edge’) between two nodes represents a probabilistic dependency between the linked nodes. The links are shown with an arrow pointing from the causal node(s) ($Y_{1},Y_{2}$ in Figure 3) to the effect node (*X*: Lunge cancer in Figure 3). There must not be any directed cycles: one cannot return to a node simply by following a series of directed links. Nodes without a child node are called leaf nodes, nodes without a parent node are called root nodes ($Y_{1},Y_{2}$), and nodes with parent and child nodes are called intermediate nodes. In other words, a BN represents dependence and conditional independence relationships among the nodes using joint probability distributions, with an ability to incorporate human oriented qualitative inputs. The method is well established for representing cause–effect relationships.

BN learning consists of two general steps: (i) Finding DAG, which illustrates the interdependency between the variables/nodes and (ii) Finding CPT for each node given the values of its parents on the learned DAG. Finding the best DAG is the crucial step in BN design. The construction of a graph to describe a BN is commonly achieved based on probabilistic methods, which utilise databases of records [48] such as the search and score approach. In this approach, a search through the space of possible DAGs is performed to find the best DAG. The number of DAGs, f(p), as a function of the number of nodes, *p*, grows exponentially with *p* [49].

In this paper, BN will be used to evaluate the effect of flood interventions upon mental health to explore and display causal and complex relationships between key factors and final outcomes in a straightforward and understandable manner. The proposed BN is also used to calculate the effectiveness of the interventions and the uncertainties associated with these causal relationships, which will be discussed in the next section. Due to the lack of data, the proposed BN in this study was learned based on expert judgments (including experts from EA and Public Health of England (PHE)), and narrative in the relevant literature (as discussed in the next section). However, this approach will effectively work with data from a variety of sources, and handles a mix of subjective and objective data that can be incorporated with variables from different contexts [14,48]. Moreover, BN is a reasonable supplement to traditional statistical methods, since traditional statistical methods were unable to update complex systems in the light of new information, while BNs can update the system when new evidence is added during analysis. The proposed BN developed an understanding of the effect of flood interventions, and the risk factors associated with a higher impact on mental health outcomes.

To construct a BN for the evaluation of the effect of flood interventions upon mental health, the following steps need to be performed:BN structure learning: There are a number of risk factors related to the flood interventions upon mental health including healthcare resources, flood management practices, existing mental disorders and many more, which will be considered as input and mediate nodes in the proposed BN model. The level of effectiveness of these nodes and the causal relationships between them are presented by edges, which can be elicited from the domain experts and the available data to construct the BN structure.Parameter learning: prior probabilities assigned to root nodes and conditional probabilities for dependent (leaf) nodes are elicited from the experts’ domain and existing information. In the BN, the state of some nodes could be influenced by their prior states, or affect other nodes. The probabilities of these nodes are determined before propagating evidence to the model [50,51].Outcomes of BN (Posterior probability learning): The final step in BN is to run the model at agreed intervals. As new information is added to the model, the current priors/states will be updated using the Bayesian paradigm in a very efficient way.

It is also straightforward to use the BN to identify which variables have the largest influence on the final outcomes of the network. A unique feature of BNs is the ability to back propagate the model’s conditional probabilities through the model structure. This means that we can test how to achieve desired outcomes by identifying the most likely combination of risk factors.

The BN model can be used to develop an effective and efficient decision support tool. In the next section, the BN-based decision support tool will be developed to evaluate the cost-effectiveness (in terms of monetary value) of various flood interventions upon mental health in the presence of different uncertainties and under certain constraints.

## 7. Results

### 7.1. Using Bayesian Network to Evaluate the Effect of Flood Interventions upon Mental Health

In this section, we evaluate the impact of the flood interventions on the mental health of people affected by flooding using a BN trained by a combination of the data extracted from a narrative in the relevant literature from the published reports and expert judgments (including experts from EA and PHE). However, it is very straightforward to train a BN based on a combination of the heterogeneous data collected from surveys and other methods [14].

We first need to learn the BN for a subset of the risk factors selected in relation to the flood intervention’s impact upon mental health, including the prevalence of probable depression in people who have been flooded (Flood), loss of sentimental items (LSOI), prevalence of loss of sentimental item as secondary stressor in those exposed to flooding (PPD), less severe depression (Lsever), and more severe depression (Msever).

It is usually recommended that the BN structure and model parameters should be learned from the combination of data and expert judgments [48]. However, Vepa et al. [14] argue that the best BN structure learns from data only and, by employing various score-based or constraints-based methods [49], would not result in a model favoured by the domain experts. As a result, the BN structure illustrated in Figure 4 is learned based on the expert opinions only (as suggested in [14,48]). The learned BN for the selected risk factors was validated by the domain expert (Economic Evidence expert from the English Environment Agency). It should be noted that “CU” and “CQALY” in the learned BN shown in Figure 4 stand for “change in utility” and “change in QALY”, respectively, which will be discussed later in this section.

In the next step, we need to estimate or determine the CPTs. Due to the lack of data, the conditional probabilities of each node of the BN shown in Figure 4 are determined using the expert opinions [52] and the information extracted from the narrative of the literature. Table 1 shows the summary of the probabilities of each node, illustrated in Figure 4, and the source of information used to determine these probabilities. For instance, the probability of LOSI is reported to be 62% based on opinions elicited from the EA and PHE experts, while the prevalence of probable depression in people who have been flooded (denoted by “Flood”) is determined to be 20.1% [53] from the English National Study of Flooding and Health (denoted by NSFH2020 hereafter), which is published in 2020 and available at https://bit.ly/3eXiKwt . The probability of PPD was determined to be 18.6% [53]. The probabilities of Lsever and Msever are respectively determined to be 48.3% and 21.1% (NSFH2020).

Following [54], the three mental health states in this study were remission, less severe depression (Lsever), and increased or more severe depression (Msever). The utility value of being in remission from depression was suggested to be 0.85, while having less severe depression is 0.60 and more severe depression is 0.42. For the sake of simplicity at this stage, we assume these utility states are monitored over one year, and that remission from depression is equivalent to not having depression. Moreover, there could be some overlap between the two states of Lsever and Msever, which then need the change in utility to be computed as illustrated in Table 2. It should be noted that the mean values reported in Table 2 are computed as the means of the suggested Beta distribution (denoted by Be(α,β) in the fourth column). These Beta distributions can be used to determine the cost-effectiveness intervention by optimising the Expected Value of Perfect Information measures [55,56], which is beyond the scope of this article and will be considered as the further development of this study.

Next, the proposed BN presented in Figure 4 computes the change in QALY (CQALY) caused by loss of sentimental items (Table 3). A QALY is a measure that combines health-related quality of life and length of life into a single measure of health gain. The National Institute for Health and Clinical Excellence (NICE) provides the cost-effectiveness threshold range, which is between £20,000 and £30,000 per QALY [57,58].

Let us assume that before taking an intervention (e.g., using flood early warning system by the local EA to inform the people in advance about flood hazard) that could lead to an individual losing their sentimental item, the changes in QALY for two mental health states (i.e., Msever and Lsever) are computed from the learned BN illustrated in Figure 4 as follows:For Msever: CQALY = 0.055;For Lsever: CQALY = 0.062.

The above CQALYs are computed based on the mean values suggested for the mental states as given in Table 2.

The QALY values, if the intervention was decided to be taken by the local EA prior to the flooding, will be computed (from the BN) as follows:For Msever: CQALY = 0.033;For Lsever: CQALY = 0.038.

The differences that the intervention will make to the mentioned mental health states are given by

For Msever: 0.055 − 0.033 = 0.022;For Lsever: 0.062 − 0.038 = 0.024.

Finally, by multiplying these changes in QALY by the lowest point of NICE’s QALY cost-effectiveness threshold (e.g., £20,000), we can evaluate the cost-effectiveness of the suggested intervention on reducing the mental health impact due to the loss of sentimental items, as:For Msever: 0.022×£20,000 = £440;For Lsever: 0.024×£20,000 = £480.

This suggests that using a flood early warning system from the local EA to inform people could save at least £480 to ensure that an individual will not suffer the less severe depression due to losing their sentimental items in flooding events.

It should be noted that the early warning system could itself create further stress. An alternative way would be to relocate people away from the affected regions as soon as possible, and to support them during and after a flood. However, any of these strategies or their combinations could affect the flooded people’s mental health, with each strategy imposing varying benefits and costs. The method proposed above can provide us with an effective cost–benefit analysis approach in comparing the suggested interventions, by taking into account the complex relationships between flood events, their aftermath, population wellbeing and risk factors causing people’s health deterioration and/or psychological disorders, and the costs and benefits of the interventions.

## 8. Conclusions

BNs have been written to evaluate (ex-post) the effect of different factors on outcomes, in contexts other than flooding. For instance, a BN has represented the interactions of indoor climate factors on the mental performance of office workers, to demonstrate that investment in improved thermal conditions is economically justified in most cases with different building designs [44]. This paper is interesting because it is not an evaluation of a single case, but attempts to gain insights across a diversity of past cases. Clearly, if a BN accurately reflects the conditional probabilities of past cases, it can be used for assessment (ex-ante) and quantification of forthcoming designs of buildings, but there are considerations to be addressed.

A BN used for evaluation, based on retrospective data and expert opinion, would need to change to represent the future scenario being assessed. The scenario may introduce new variables (and associated changes to prior probabilities), for example, to represent how the future complexity of the system will work. The scenario may require changes to the values of existing variables, for example, to reflect future efficiencies or the effects of different ways of organising. In fact, a number of BNs may be developed to examine alternative viable futures. Every BN will have a set of unique conditional probabilities which will assess the future conditions or scenarios. Across all BNs, a range of potential outcomes will ensue and will provide an indication of future outcomes. However, scenarios are speculative and indeed deterministic [59], but if they extend current representations of factors upon outcomes, they may be argued as rational extensions of current understanding. In general, a BN-based modelling approach would enable us to compute the costs and benefits, based on multiple causal factors including individual risk factors or interventions on the system, by taking into consideration uncertainty in the input parameters. The derived estimated outcomes could be presented in the form of probabilities if an appropriate probability distribution could be considered for the input parameters (see Section 7.1). It should be noted that the BN, similar to many other machine learning methods, is a data-driven approach. As a result, the results derived using BN are not generalisable and are fully dependent on the collected data and the assumption that the data sources are accurate. We identify this as a limitation of the proposed approach.

The health economic evaluation methodology proposed in this study, to explore the level of investment required for alternative interventions based on desired mental health outcomes, could be developed further by computing the value of information (VoI) analysis [55] (e.g., the expected value of perfect information or EVPI). The cost-effectiveness of the suggested interventions for flood morbidity related mental health depression should be compared in terms of both the healthcare perspective and the societal perspective, using the evaluation method proposed in this study and a VoI analysis, which estimates the expected value of eliminating the uncertainty surrounding cost-effectiveness estimates, for both perspectives. Furthermore, the concept of the expected value of perfect information, which is a particular measure of VoI analysis, can be used to examine probabilistic sensitivity analysis for the discussed cost-effectiveness problem (see [55,56]).

It would also be interesting to further explore the price or threshold that a healthcare decision maker or policymaker would be willing to pay/meet to have perfect information regarding all factors that influence which intervention choice is preferred as the result of a cost-effectiveness analysis. However, this could be answered by VoI analysis to find the value (in monetary terms) after removing all uncertainty from such an analysis, more research and data are required to investigate the effectiveness of the proposed method in this regard.

## Figures and Tables

**Figure 1 ijerph-18-07467-f001:**
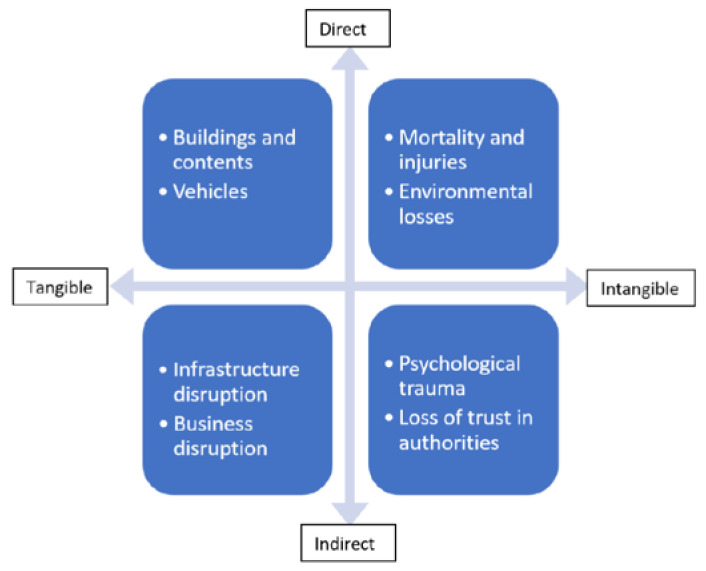
There are direct and indirect impacts of flood damages that are not easily quantified in monetary terms [6].

**Figure 2 ijerph-18-07467-f002:**
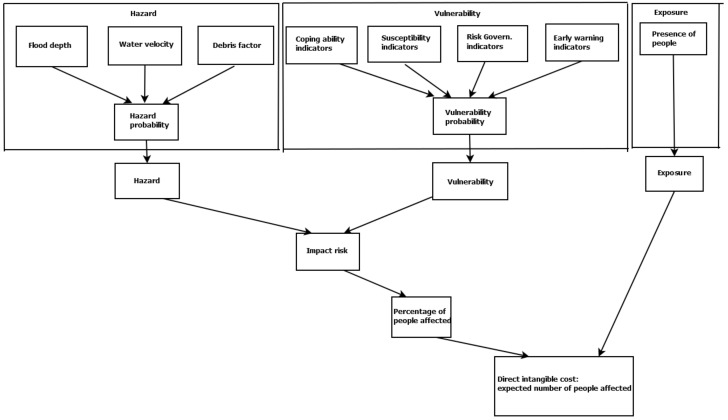
Customised application of the risk framework by including an early warning system (some information derived from the original framework developed by [33]).

**Figure 3 ijerph-18-07467-f003:**
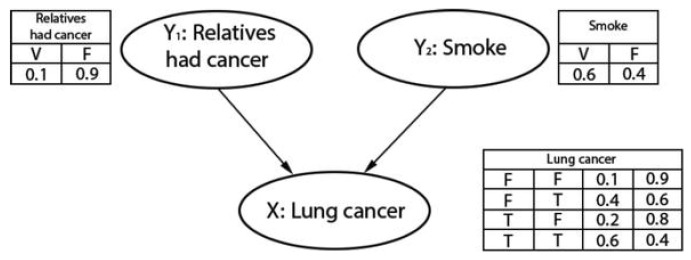
A simple BN model indicating the inter-dependencies between a lung cancer classifier and the affecting risk factors (adopted from [47]).

**Figure 4 ijerph-18-07467-f004:**
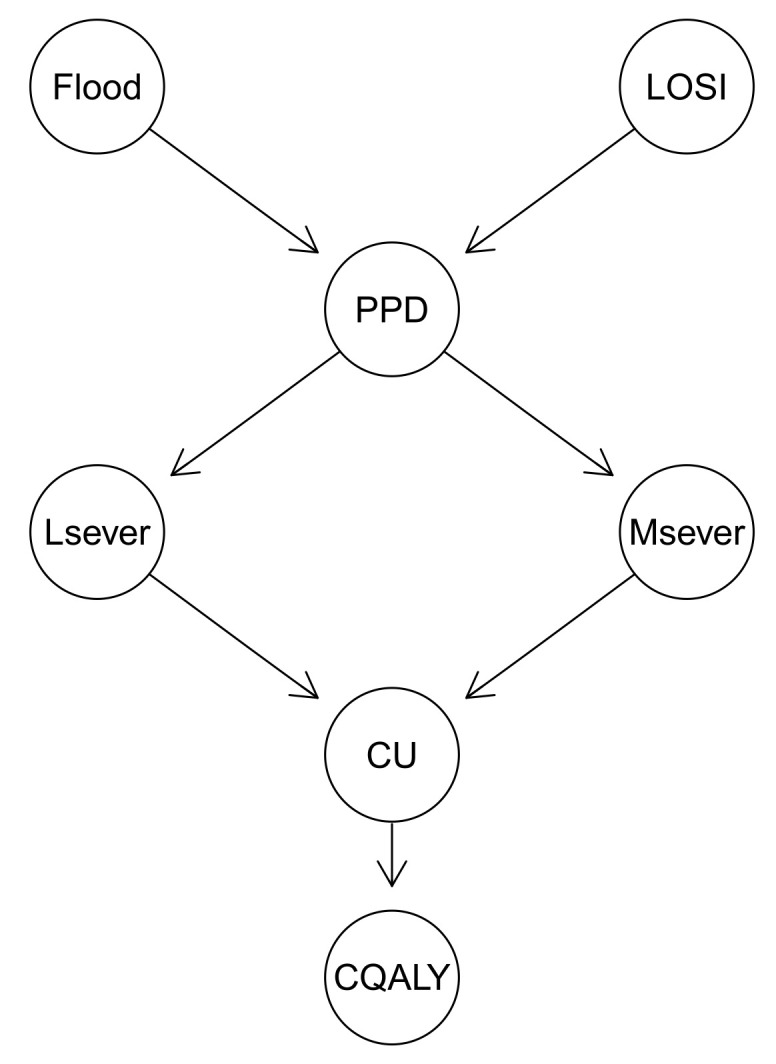
Static Bayesian network to evaluate the effect of flood interventions upon mental health where LOSI indicates the loss of sentimental items, PPD indicates the prevalence of probable depression, Msever and Lsever indicate the more severe depression and less severe depression respectively, CU indicates the change in utility and CQALY indicates the change in QALY.

**Table 1 ijerph-18-07467-t001:** The elicited probabilities and corresponding source of data for each node of BN illustrated in Figure 4.

Input Parameter (Node)	Probability	Source of Data
Flood	20.1%	[53] (p. 8)
LOSI	62%	domain experts’ opinions
PPD	18.6%	[53] (p. 15)
Lsever	48.3%	NSFH2020
Msever	21.1%	NSFH2020

**Table 2 ijerph-18-07467-t002:** The states of mental health and their corresponding utility values, as suggested in [54].

Input Parameter (Health States)	Mean Value	Change in Utility	Probability Distribution	Source of Data
Remission	0.85		Be(923,163)	[54]
Lsever	0.60	(0.85−0.6)=0.25	Be(182,122)	[54]
Msever	0.42	(0.85−0.42)=0.43	Be(54,75)	[54]

**Table 3 ijerph-18-07467-t003:** The change in QALY outcomes due to an intervention taken by the EA.

Health State	Before Intervention	After Intervention	The Difference	CQALY Outcomes
Msever	0.055	0.033	0.022	0.022×£20,000 = £440
Lsever	0.062	0.038	0.024	0.024×£20,000 = £480

## Data Availability

Due to the lack of the relevant secondary data, the machine learning model proposed in this study was learned based on the domain expert opinions and the narrative of the relevant literature, such as the English National Study of Flooding and Health (NSFH2020), which is published in 2020 and available at https://bit.ly/3eXiKwt.

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
