# Peer review of "Economic Evaluation of Mental Health Effects of Flooding Using Bayesian Networks"

_ijerph, 2021, doi:10.3390/ijerph18147467_

Round 1

Reviewer 1 Report

The author conducted an economic evaluation of mental health effects of flooding using Bayesian Networks. This is an interesting study for identifying social value for investing interventions of natural disaster victims (flooding). I think this study might follow the logical process for identifying conditional relationship between the cause and health results using DAGs, and estimating reasonable (breakeven) point of for investing mental health interventions based on QALYs of victims and national threshold for NHS in UK. I think some vague points should be addressed to get the reader understand, which are as follows. Compulsory Revision 1. Please describe the methods separately, and include the network analysis and utility and cost estimation methods and other economic evaluation process (sensitivity analysis) in this method parts. 2. Please add result part and describe cost and QALYs identifying and estimating results with tables separately in this result part. 3. The author should describe more details about the QALYs calculation process (line 403). Utility estimations, time horizon and their confidence (credible) intervals, sources, and QALYs calculation method should also be described and added the calculation process as table form. 4. Along with deterministic and subgroup (depression severity) analysis results, some scenario analysis (using geographic, demographic or meteorological scenarios from ex ante perspective) and probabilistic sensitivity analysis (using distribution and critical values) need to be added for understating the uncertainty of this study results. Finally, in the discussion, the author needs to describe some limitations of this study.

Author Response

We would like to thank the esteemed reviewers for taking the time in reading our manuscript. We also believe that the points in your feedback were very helpful and were certainly beneficial to the quality of our work; therefore, there is no rebuttal on the points raised. In places that we felt the review point is due to our unclear stance, we have attempted to provide more clarity within the work.

Please see the attached file our responses to Reviewer 1 comments. 

Reviewer 2 Report

I do believe it is an interesting paper. I have the following suggestions:

  1. Can the author compare to other method?
  2. The BN section need to provide more details.

Author Response

We would like to thank the esteemed reviewers for taking the time in reading our manuscript. We also believe that the points in your feedback were very helpful and were certainly beneficial to the quality of our work; therefore, there is no rebuttal on the points raised. In places that we felt the review point is due to our unclear stance, we have attempted to provide more clarity within the work.

Please see the attached file our responses to Reviewer 2 comments. 

Reviewer 3 Report

Dear Authors;

The manuscript has some serious issues as follows. Given these, I recommend a major revision and resubmission.

P.S:

[1] Structure:   When a new method is offered to improve estimations, applying it to few real-life examples is not enough. The associated results are not generalizable. The authors need to add one section in the results section with the simulated data and present its associated results.

[2] References: It is appropriate as the paper text goes on, the references to be added chronologically. The current order of references needs to be rearranged. 

[3] Abbreviation list:  It is appropriate for the paper to have the list of used abbreviations and their long format before the Reference section.

Author Response

We would like to thank the esteemed reviewers for taking the time in reading our manuscript. We also believe that the points in your feedback were very helpful and were certainly beneficial to the quality of our work; therefore, there is no rebuttal on the points raised. In places that we felt the review point is due to our unclear stance, we have attempted to provide more clarity within the work.

Please see the attached file our responses to Reviewer 3 comments. 

Reviewer 4 Report

“This paper contributes a design for a Bayesian network that exposes causal pathways and conditional probabilities between interventions and mental health outcomes as well as providing a tool which can readily indicate the level of investment needed in alternative interventions based on desired mental health outcomes.”

  1. The Data and Methodology sections should be provided as separate. The selection of priors should be given more discussion.
  2. There is no comparison of the obtained outcomes with some real data.
  3. The estimated model is not compared with any alternative model.
  4. There is no discussion of any kind of prediction possible to get from the model, which would be compared with some forecast accuracy measures (and tested) with some real data.

The whole paper is interesting and seems to be free of important flaws or mistakes. However, the core of the research in the current state seems to be nothing more than a simple illustration of what a Bayesian network is and how it can be constructed. It could be included as some Example in a didactic material, but, for me, seems to be too little for a separate research article. Even some post-conference publications in this topic seem to contain a bit more deeper analysis of the described cases.

Author Response

We would like to thank the esteemed reviewers for taking the time in reading our manuscript. We also believe that the points in your feedback were very helpful and were certainly beneficial to the quality of our work; therefore, there is no rebuttal on the points raised. In places that we felt the review point is due to our unclear stance, we have attempted to provide more clarity within the work.

Please see the attached file our responses to Reviewer 4 comments. 

Round 2

Reviewer 1 Report

The changes made by the authors to the original version improved the transparency, clarity and methodological consistency of the manuscript considerably. The comments I raised have been addressed, adequately. Therefore, I recommend this article for publication in the Int. J. Environ. Res. Public Health after the collection and consideration (not compulsory) listed below.

Correction: line 490 “sever” -> “severe”

Consideration: In line 489, the author mentioned “This suggests that using flood early warning system by the local EA to inform the people could save at least £480 to ensure that an individual will not suffer the less severe depression due to losing their sentimental items in the flooding events”. I don’t know whether the victim’s QALY loss caused by fearfulness against flood itself or PTSD after losses from flood. Therefore I think that the early warning system might not fully protect their less severe depression due to losing their sentimental items.

Author Response

We would like to thank the esteemed reviewer for taking the time in reading our revised manuscript. Thank you for acknowledging strengths in the revised manuscript and recommending it for the submission.

We have corrected “sever” by “severe” in line 490. Thank you for pointing out this. In order to respond to your "Consideration point", we have added the following description underneath line 491. 

It should be noted that the early warning system could itself create further stress. An alternative way would be to relocate people away from the affected regions as soon as possible, and to support them during and after a flood. Although, any of these strategies or their combinations could affect the flooded people’s mental health, with each strategy imposing varying benefits and costs. The method proposed above can provide us with an effective cost-benefit analysis approach in comparing the suggested interventions, by taking into account the complex relationships between flood events, their aftermath, population wellbeing and risk factors causing people’s health deterioration and/or psychological disorders, and costs and benefits of the interventions.

Reviewer 2 Report

I thank the authors for responding to each of my comments.

Author Response

We would like to thank the esteemed reviewer for taking the time in reading our revised manuscript. Thank you for acknowledging strengths in the revised manuscript and recommending it for the submission. 

Reviewer 3 Report

Dear Authors, most of the raised points were addressed. Regards,

Author Response

(The authors gave the same response as above.)

Reviewer 4 Report

I still cannot see any point in publishing a paper that adapts well-known and relativelly simple technique but without any verification on real data. For me, such a paper is not a paper, but just a draft of a plan of the future research. In other words, until real data will be inserted, this remains for me as a sketch or plan of a research - not a full and finished research. On the other hand, a collection of various ideas can be made into a review paper; but this paper is neither a review one.

Which kind of usuability comes from this paper? How does the numbers obtained in this paper relate to real-life cases?

Author Response

We would like to thank reviewer 4 for his/her on-going challenge. We believe that we have demonstrated the use of real data, i.e. data from domain expert opinions and the narrative of relevant literature, such as the English National Study of Flooding and Health (NSFH, 2020) available at https://bit.ly/3eXiKwt. This is real albeit not perhaps the type of data that the reviewer is demanding. We argue that our data is stronger, because it takes into account a broader perspective on the understanding of the range of connections between policy interventions and mental health outcomes. Real (empirical) data would be idiosyncratic and lead to only partial inference on this research topic. We have stated already that there is insufficient data collected on the whole topic, and so generalisation would not be possible (please see our discussion in the 2nd paragraph of Conclusions).  We have added further text to explain this in lines 416-500, including the detailed information about the sources of collected data, provided in Tables 1 and 2. Thank you again for your feedback.